# Gene Therapy Targeting p53 and KRAS for Colorectal Cancer Treatment: A Myth or the Way Forward?

**DOI:** 10.3390/ijms222111941

**Published:** 2021-11-03

**Authors:** Hidayati Husainy Hasbullah, Marahaini Musa

**Affiliations:** Human Genome Centre, School of Medical Sciences, Universiti Sains Malaysia, Kubang Kerian 16150, Kelantan, Malaysia; hidayatihh@student.usm.my

**Keywords:** colon cancer, mutation, p53, KRAS, targeted therapy

## Abstract

Colorectal cancer (CRC) is the third most commonly diagnosed malignancy worldwide and is responsible as one of the main causes of mortality in both men and women. Despite massive efforts to raise public awareness on early screening and significant advancements in the treatment for CRC, the majority of cases are still being diagnosed at the advanced stage. This contributes to low survivability due to this cancer. CRC patients present various genetic changes and epigenetic modifications. The most common genetic alterations associated with CRC are p53 and KRAS mutations. Gene therapy targeting defect genes such as *TP53* (tumor suppressor gene encodes for p53) and *KRAS* (oncogene) in CRC potentially serves as an alternative treatment avenue for the disease in addition to the standard therapy. For the last decade, significant developments have been seen in gene therapy for translational purposes in treating various cancers. This includes the development of vectors as delivery vehicles. Despite the optimism revolving around targeted gene therapy for cancer treatment, it also has various limitations, such as a lack of availability of related technology, high cost of the involved procedures, and ethical issues. This article will provide a review on the potentials and challenges of gene therapy targeting p53 and KRAS for the treatment of CRC.

## 1. Introduction

Cancer is one of the major health issues affecting men and women globally. It is estimated that 19.3 million new cancer cases and nearly 10.0 million cancer deaths occurred in 2020. Colorectal cancer (CRC) was recorded as the third most commonly diagnosed malignancy worldwide, following breast and lung cancers. CRC qualified as the second leading cause of cancer death after lung cancer. Developed countries possess the highest risk of colon and rectal cancer [1].

Surgical resection and conventional therapy such as chemotherapy have long since been considered as the ‘gold standard’ management for CRC [2]. Despite many initiatives to increase public awareness for early screening of CRC and advancements in cancer therapy, the majority of cases are diagnosed at the advanced stage, which contribute to the poor prognosis of this cancer [3,4].

Cancer arises from the accumulation of somatic mutations and other genetic changes which affect cell division checkpoints and result in both abnormal cell growth and eventually tumorigenesis. The process of carcinogenesis occurs, in which abnormal cells acquire advantageous biological capabilities known as the hallmarks of cancer [5]. Similar to other types of cancer, genetic and epigenetic changes are implicated in the progression of CRC. Mutations of *TP53* (tumor suppressor) and *KRAS* (oncogene) are among the most commonly reported genetic alterations in CRC [6].

Conventional treatments such as chemotherapy impose serious adverse side effects on patients [7]. It is imperative that safer and more effective options be included as part of cancer management. One of the proposed therapies for CRC is targeted gene therapy. This approach aims to improve the life expectancy of the patients and reduce the risk of disease recurrence. Advancement in the field of gene therapy offers promise to various innovative treatments that are likely to become prominent in improving the survivability of cancer patients. Gene transfer is one of the treatment modalities that introduces new genes into a malignant cell or into the surrounding tissue to induce cell death or slow cancer proliferation. Flexibility of this technique and the ability to utilize a wide range of genes and vectors have solidified its successful outcomes in many clinical trials. As these gene therapy technologies in oncology mature, they may be used independently or in combination with current or conventional treatments for cancer management [8,9].

## 2. Colorectal Carcinogenesis

CRC is a cancer that originated from the colon and rectum. It starts with transformation from a normal colonic crypt to benign intestinal polyps. A small number of these (adenomatous polyps) will progress into advanced adenoma and become cancerous growths (malignant neoplasia) and metastases. The majority of polyps such as inflammatory and hyperplastic polyps are benign and not categorized as pre-malignant lesions [10]. Most CRCs are adenocarcinomas. The gradual transition occurs in CRC development, also referred to as ‘multistep tumorigenesis’, and specific genetic changes in tumor suppressors or oncogenes are associated with every stage [11,12,13].

CRC can be classified into four groups based on their consensus molecular subtypes (CMSs), namely [14];
(a)CMS1: microsatellite instability (MSI) and strong activation of immune components(b)CMS2: exhibits epithelial differentiation and upregulation of the wingless-related integration site (WNT) and MYC signaling;(c)CMS3: prominent metabolic dysregulation; and(d)CMS4: evident expression of genes compatible with stromal invasion, epithelial–mesenchymal transition (EMT), transforming growth factor-beta (TGF-β), and angiogenesis.


A small fraction of mutations, referred to as ‘drivers’, are responsible for the cellular transformations that result in cancer formation. Driver mutations characterize molecular phenotypes of tumors and act as predictors in clinical outcomes for patients [15]. Cells harboring driver mutations acquire selective advantage, leading to their proliferation, and these mutations have been positively selected during cancer evolution. They are considered as a part of the ‘cancer genes’. Other than driver mutations, our cells also accumulate other mutations referred to as ‘passenger’ mutations, which may not seemingly affect the cancer progression. Despite this, passenger mutations attract more interest as it can impact epigenetics, immunogenicity, or the response to therapy for cancer patients [16]. Three key players in promoting CRC progression are tumor suppressor genes, oncogenes, and mismatch-repair genes. Mutations in those genes may occur during the gestational period or may be acquired over the lifetime (inherited versus somatic mutation). Although many scientists have focused more on the roles of somatic mutation rather than germline mutation in the predisposition to cancer, it is undeniable that the hereditary aspect of mutations (germline mutation) possesses an essential role in cancer development and metastasis, which presents clinical applications in cancer studies. CRC represents one of the largest percentages of familial cases [17,18]. It is also worthy to note that the majority of CRC cases (70%) arise from sporadic mutation and are significantly influenced by external risk factors such as lifestyle, lack of physical activity, and unhealthy diet [4,17]. 

It is an established notion that CRC arises from accumulations of genetic and epigenetic changes. These modifications promote the formation of carcinoma from normal mucosa by altering the signaling pathways involved in the regulation of cancer behaviors [19,20]. Six key driver genes in CRC, namely adenomatous polyposis coli (*APC*), *KRAS*, *BRAF*, *PIK3CA*, *SMAD4*, and *TP53*, were identified by using bioinformatics tools [21]. Figure 1 illustrates the colorectal progression and respective mutations that are involved [10,22,23,24].

Three major molecular pathways have been identified in CRC carcinogenesis: chromosome instability (CIN), as well as the MSI and CpG island methylation pathways (CIMP) [25]. In up to 80% of sporadic CRC cases, the tumor arises from the CIN pathway and progresses through the classic adenoma–carcinoma sequence. This pathway is triggered by multistep mutational events involving multiple genes, such as *KRAS* oncogene, and tumor suppressor genes, such as *APC* and *TP53*. *APC* is found to be mutated in the majority (80–90%) of inherited and sporadic CRC. *APC* mutations disrupt the *APC* complex formation, thus resulting in the b-vatenin translocation to the nucleus. This subsequently induces overactivation of Wnt-signalling and drives both cell proliferation and activation [12,26,27].

CRC formation also involves a dynamic interaction between malignant cells, components of the microenvironment (stromal and vascular endothelial cells), and the immune system [28,29,30]. Increased activation of fibroblasts occurs alongside CRC progression. Accumulation of cancer-associated fibroblasts (CAFs), which is a major component of the tumor microenvironment, contributes to poor prognosis and recurrence of CRC [31]. These stromal cells support cancer cell growth and differentiation via secretion of cytokine, and vice-versa [32]. It is worth noting that variation in the genetic makeup also (i.e., polymorphism) explains the differences in responsiveness to infection and regarding the risk of developing CRC between different populations or races [33,34].

## 3. The p53 and KRAS Mutations in CRC

Cancer profiling can be performed by analyzing the genetic alterations or changes that contribute to its carcinogenesis. Table 1 summarizes main genetic mutations detected in several of the most commonly diagnosed solid tumors [35,36,37,38,39]. The listed genes serve as possible targets for designing targeted therapy against different cancer types. We note that p53 and KRAS mutations are not only reported to be prevalent in CRC but also in many other cancers, such as malignancy of the breast, lung, and pancreas. 

Although Table 1 lists the common mutations involved in solid tumors, the frequency and mutational profiling of cancer patients may differ greatly depending on their genetic makeup and tumor heterogeneity. Taking *TP53* as an example, mutation sites of this gene are diverse between cancer subjects. Thus, it is important to analyze different tumors thoroughly in order to cater to more personalized therapy based on patients’ mutational profiles. 

The Cancer Genome Atlas Network reported the most frequently mutated genes in the CIN pathway, which include *APC*, *TP53*, *KRAS*, *PIK3CA*, *FBXW7*, *SMAD4*, *TCF7L2* and *NRAS* [40]. One of these genes, namely *TP53*, is the second most frequent gene to be mutated in colorectal cancer, affecting approximately 60% of CRC patients. The *TP53* gene, located at the short arm of chromosome 17, encodes the important tumor suppressor protein p53. p53 acts as a transcription factor that regulates cell cycles and apoptosis by binding to specific sequences at various genes including p21, Bax, and Bcl-2 in response to cellular stressors such as DNA damage or hypoxia [41]. Activated p53 via different target molecules can (a) induce temporary cell cycle arrest to allow for repair of DNA damage; (b) induce apoptosis; and (c) abolish further cell division [42]. Loss of p53 transcriptional activity therefore results in unregulated cell growth and promoted tumorigenesis in various organs including the colon. In the majority of cases, the *TP53* mutations in CRC cancer cells are missense-type mutations in the DNA-binding domain [41]. This mutation produces mutant p53 with a reduced capacity to bind to the specific sequence that regulates the p53 transcriptional pathway. In approximately 70% of CRC tumors that carry the missense mutations, the wild type *TP53* is lost via loss of heterozygosity (LOH) due to chromosomal instability [25]. It is proposed that the missense-type mutations of p53 drive the progression of adenoma to adenocarcinoma and the additional loss of wild type *TP53* results in a more advanced progression of CRC [41]. 

Among the genes of interest stated by the Cancer Genome Atlas Network (2012) [40], *KRAS* gene mutation testing is the most useful assessment, routinely used currently to predict response to anti-EGFR therapy. KRAS (Kirsten ras sarcoma viral oncogene) is a guanosine triphosphate (GTP)-binding protein located downstream from EGFR in the KRAS-BRAF-MEK-ERK pathway, which is one of the most important mitogen-activated protein kinase (MAPK)-signaling pathways. The ERK-MAPK pathway is important in cell proliferation and differentiation. Mutations in *KRAS* result in EGFR-independent activation of the MAPK pathway by reducing the GTPase activity that leads to an increase of cell proliferation potentially occurring in the early stage of colorectal carcinogenesis [43]. It is also proposed that the ERK-MAPK pathway is involved in the progression and oncogenic behavior of human CRC.

Various studies have demonstrated that mutations in the *KRAS* are implicated in approximately 30–40% of CRC patients [44,45,46]. The proportion of *KRAS* mutations differed between regions and races. For example, in Malaysia, the prevalence is lower (approximately 20%), as previously reported [47,48,49]. More than 90% of the activating mutations occurred at codon 12 and codon 13 in exon 1, with single-base substitution of glycine to aspartate (G12D) and glycine to valine (G12V) on codon 12 being the most common mutations observed [45]. Mutations at codons 61 and 146 have also been recorded, although these account for only about 1–4% of *KRAS* mutations and its clinical significance remain unclear [50]. 

Associations between the *KRAS* mutational status and clinicopathological features of CRC have been studied but there is no consensus obtained regarding its relation. Hayama et al. (2019) and Scott et al. (2020) [42,46], for instance, demonstrated that *KRAS*-mutated CRC were more frequently observed at the right side of the colon, while Zulhabri et al. (2012) [48] reported that these mutations are more common on left-sided CRC. Other studies including the collaborative RASCAL study [51] and study by Phipps et al. (2013) [52] showed no association between *KRAS* mutations and tumor site. Zulhabri et al. (2012) also explored the relationship between different ethnicities and *KRAS* mutational profiles but found no significant association between the variables [48]. The missense-type p53 mutation expression or suppression of TGF-β induces submucosal invasion. Additional *KRAS* gene mutation promotes more malignant transformation (epithelial–mesenchymal transition and metastasis) [53]. 

Molecular characterization of cancer-associated mutations over the more recent years has been proven to provide essential information on disease prognosis and response to treatment such as chemotherapy. Two previously unreported prognostic associations in CRC, namely the *TP53* mutation and the total mutation burden, confirmed correlations with the *KRAS* homolog and B-Raf proto-oncogenes (*BRAF*) were identified through multigene panels. This finding has provided insight into the importance of information acquired from modest-sized genetic and molecular testing in clinical practice [54]. This is supported by a more recent study, in which it was reported that multiple variables, such as molecular markers (e.g., *KRAS* mutations), tumor location with the other clinical-pathological variables, and microsatellite status, could be used in combination to determine the prognosis and act as predictive factors in stage I-III CRC [55]. Another study by Tortola et al. (2016) also reported the correlation between *TP53* mutations alone or in combination with *KRAS* mutations, with poor outcomes in CRC patients, although they did not recommend the routine use of these mutations as prognostic markers in clinics [6]. 

*KRAS* mutations are associated with more aggressive tumor phenotypes and poor prognosis in CRC patients in comparison to those with wild type-KRAS [56,57]. Mutations in codon 12 and 13 of KRAS, BRAF, and phosphatidylinositol-4,5-bisphosphate 3-kinase catalytic subunit alpha (PIK3CA) also act as a predictor for resistance to anti-EGFR therapies. Thus, it has been proposed that these patients with *KRAS* or *NRAS* mutations should not be treated with therapies targeting EGFR. Instead, a BRAF inhibitor should be included in the anti-EGFR therapy for patients with BRAF mutations to overcome treatment resistance [58,59]. 

## 4. Gene Therapy for Cancer

Management of CRC involves (a) surgery; (b) conventional chemotherapy; (c) immunotherapy; and d) various gene therapy technologies such as gene correction and virus-directed enzyme–prodrug therapy [10]. Gene therapy represents a promising approach as it allows for a more personalized treatment. This would ensure the success of treatment clinically. In brief, gene therapy can be referred to as the delivery of specific genetic material (genes, gene segments, or oligonucleotides) to change the encoding of a gene product or to alter the biological properties of the host cells for the treatment of various pathological conditions or diseases [60]. Gene transfer therapies are performed either through *in vivo* or *ex vivo* methods. 

Viral and non-viral delivery vehicles are used to administer drugs or genes of interest depending on the desired specificity of the gene therapy. Targeted gene therapy is preferred over the recombinant therapeutic use of peptides, where the latter is usually characterized by low availability, lack of stability, a significant toxicity effect, and expensive cost [61]. 

Targeted gene therapy represents a novel treatment avenue to manage a disease with no definitive cure. There has been good progress in gene therapy for cancer in the last three decades, wherein a number of drugs have been approved and many are still at the stage of clinical trials. In the future, tumor genomic profiling coupled with the analysis of host humoral and cellular immunity will ensure the selection of the most appropriate patient and treatment modalities for gene therapy [62].

Gene therapy serves as a treatment avenue that can be used in combination with existing and conventional therapies. Strategies in gene therapy usually involve the use of interventional genetic techniques aiming to a) boost the immunological response to the tumor and b) deliver cytotoxic or lethal agents, and/or genes to neoplastic cells, performed via various clinical trials [63]. These are performed through different mechanisms such as including or replacing defect genes, knocking down genes, deactivating malfunction genes, or inserting new genes, which are administered using delivery vehicles or vectors [64]. Figure 2 summarizes the vectors and strategies in gene therapy for cancer [60,65,66,67,68,69,70]. Several treatment modalities which can potentially be incorporated into gene therapy have also been proposed for cancer management, including [71]:
(a)Gene-directed enzyme prodrug therapy (GDEPT), which aims to reduce the toxicity effect after chemotherapy;(b)Cancer drug-resistance gene transfer to elevate the chemotherapy and radiation effects onto the tumor cells without compromising the normal tissues; and(c)Combination of diagnostic and therapeutic systems (theranostic).


As stated in the previous section, CRC progression is also heavily influenced by the tumor microenvironment. Besides targeting cancer cells alone, blocking the angiogenesis process is one the prime methods for CRC treatment. Targeted therapy using AAV to deliver anti-angiogenic genes, namely angiostatin and endostatin, has resulted in tumor shrinkage with little side effects [72].

According to ClinicalTrials.gov, in August 2021, there were 140 clinical trials involving CRC ((accessed on 30 August 2021) https://www.clinicaltrials.gov). This data signifies a massive interest and effort from the medical community to establish gene therapy as a key treatment modality for CRC. Different strategies have been explored in targeted cancer gene therapy for CRC using various vectors and delivery approaches *in vitro* and *in vivo*. Table 2 summarizes examples of the viral and non-viral delivery systems used for CRC gene therapy in pre-clinical and clinical stages [73,74,75,76,77,78,79,80,81,82,83,84,85,86,87,88]. It is worth noting that encouraging data on gene therapy efficacy was gathered from these studies and the majority of the stated treatment strategies produced minimal or no side effects.

## 5. Gene Therapy Targeting p53 and KRAS in CRC

The principle of targeting driver mutations for CRC treatment has intrigued the scientific community for ages. These mutations, which play a vital role in carcinogenesis, represent specific targets of a tumor and generally are expressed by the majority, if not all, neoplastic cells [89]. Gene therapy targeting p53 has been explored for various solid tumors. The p53-based gene therapy in oncology commonly aims to administer wild type-p53 or to suppress mutant p53 expression in p53-defective cancer cells [90,91]. Suppression of mutant p53 function is proposed to be a robust approach to block the malignant development of CRC [41,92]. 

The first-ever gene therapy product to treat head and neck squamous cell carcinoma (HNSCC), namely Gendicine (recombinant human p53 adenovirus), was developed by Shenzhen SiBiono GeneTech Co. Ltd. Gendicine entered the commercial market in 2004 following approval by the China Food and Drug Administration (CFDA) a year earlier. Gendicine functions through adenoviruses carrying the p53 gene. The wild type p53 protein expressed by Gendicine-transduced cells acts as a tumor suppressor, activated by cellular stress, and mediates cell-cycle arrest and DNA repair, or, depending on the cellular stress state, will promote programmed cell death (apoptosis, senescence, and/or autophagy). In short, by restoring p53 activity and thus putting a ‘brake’ to the cell division process, cancer development is subsequently blocked. The commercial success of Gendicine was evidently shown by its safety record, reporting that it has been used in more than 30,000 patients, involving no less than 30 published clinical trials. In combination with conventional treatment, namely chemotherapy and radiotherapy, Gendicine has exhibited significantly better response rates compared to the standard therapies alone, with no significant adverse effects reported [69]. In China, Gendicine has been clinically used to treat malignancies other than HNSCC for more than a decade now mainly for patients with advanced or unresectable cancerous tumors. However, standard treatment regimens utilizing p53 gene therapy still need to be established for advanced use in clinical practice [93]. This successful track record of Gendicine has cemented the idea of targeted treatment in oncology and has provided valuable insight into the potentials of gene therapy for treating malignancies.

Besides Gendicine, targeted gene therapies for cancer include Rexin-G^®^ and VB-111, which are used for the treatment of pancreatic cancer and glioblastoma, respectively. Rexin-G^®^ is the first tumor-targeted gene therapy vector-bearing cytocidal cyclin G1 construct that has been tested in the clinical setting [94,95]. VB-111 (ofranergene obadenovec), in contrast, is a non-replicating adenovirus carrying a Fas-chimera transgene (Fas and human tumor necrosis factor receptor 1) that induces targeted apoptosis of tumor angiogenic blood vessels and promotes a tumor-specific immune response [96,97]. 

The curative effect of the combination of gene therapy using the p53 gene and standard treatment in treating esophageal cancer has also been explored by Cui et al. (2015) [9]. In this study, middle to advanced stage esophageal cancer patients were treated with a combination of gene therapy, with a recombinant human adenovirus-p53 (rAd-p53) vector, and chemotherapy. Shrinkage of the tumors was reported post-injection and besides self-limiting fever, no side effects were recorded. 

A number of cancer and/or tumor-specific promoter systems have been designed to target malignant cells. This potentially highlights the potential of including the promoter for tumor-suppressor genes delivered to cancer cells using a suitable genetic vector [98]. A more recent development in CRC gene therapy is the establishment of nanoparticles (NPs) from mesoporous silica nanoparticles (MSNs) as novel genetic vehicles, which are reported to effectively bind and target CRC cells [99,100]. 

Compared to gene therapy methods involving p53, the approach of targeting KRAS is not fully developed in oncology practice. However, several studies have reported on the potential of targeted gene therapy against KRAS. In pancreatic cancer, where high prevalence of *KRAS* mutation was found (95% of the cases), overexpression of the pro-apoptotic protein, which is the p53-upregulated modulator of apoptosis (PUMA) under a Ras responsive promoter, was proved to be effective in selectively killing Ras-transformed cells *in vitro*. This method was suggested to be a useful, efficient, and safe strategy to eradicate the Ras-mutated tumor cells [101,102].

## 6. Challenges

### 6.1. Inaccessible Genetic Testing

CRC screening is usually performed using stool-based tests such as the fecal immunochemical test (FIT) and endoscopic methods (i.e., colonoscopy). Both approaches evidently lower the risk for mortality due to CRC and their benefits outweigh the limitations presented by each screening [103]. It is worthy to note that to cater to more effective targeted therapy in CRC subjects, genetic testing to identify mutational profiles, including MSI status and both p53 and KRAS mutations, is required. In many countries, the cost for mutational screening is considered expensive and may not be accessible to the majority of the population due to a lack of technologies and facilities [104]. This hinders the progress of targeted gene therapy in CRC. 

### 6.2. Variability in Response and Efficacy

A major issue with gene therapy in oncology concerns resistance to treatment with cancer recurrence and low survival rates. Tumor cell resistance following the therapy potentially is a result of an intrinsic mechanism or is acquired through the dysregulation and release of Bcl-2, which inhibits the apoptosis process [72]. This also may relate to the rate of success for gene therapy and selection of cancer patients. Many candidates enrolled in clinical trials are those with advanced stages of cancer. The efficacy of gene therapy most likely will be influenced by the severity of the disease as the cancer progresses and the complexity of both its genetic profile (mutational signatures) and immune system also increases, which can contribute to unsuccessful therapy in those subjects in comparison to having patients with earlier stages of malignancy as candidates. 

Despite the success of gene therapy using adenoviral vectors expressing wild-type p53 in many clinical trials, there are questions regarding the variability and insufficient gene delivery to every neoplastic cell, as well as regarding the possible effects of antibodies to adenovirus in humans, which may have limited its clinical application [105]. Although reintroducing wild-type p53 through gene therapy is a seemingly straightforward and principally sound approach, technical difficulties, which affect the efficacy of the gene delivery and present possible safety issues with the vector, may present as major drawbacks to this method [106]. Moreover, most of the molecules, such as inhibitors or compounds (synthetic or natural agents) used to reactivate and restore p53 function in CRC cells, were only tested *in vitro* and in animal models but not yet in clinical trials [107]. 

In terms of specificity, although *KRAS* mutation is an early driver for colorectal carcinogenesis, which makes it an attractive target for treatment, these events may not be the main initiators compared to *APC* loss or *β-catenin* mutations in MMR-deficient tumors. The extent of the dependency of these tumors on *KRAS* is still yet to be elucidated [107]. The lack of this fundamental knowledge on the KRAS pathway in CRC may have influenced the success rate of targeted gene therapy against this mutation. Several groups are studying to directly target *KRAS* mutations and thus its biological activity by either using molecules and oligonucleotides to bind the *KRAS*-mutated sites or *KRAS* synthesis inhibition at the DNA level [108,109,110]. Other proposed strategies targetting *KRAS* in CRC include using compounds to target G4 motifs present in the human *KRAS* promoter [111,112,113] and microRNAs to regulate the expression of *KRAS* or associated genes in the *KRAS*-driven pathway [114,115,116]. Although promising results were reported in inactivating mutations of *KRAS* or in the downregulation of mRNA and proteins of KRAS, and in the downstream genes in the KRAS pathway, these works are perfomed at the pre-clinical stage. Large scale randomized clinical trials are needed to verify the efficacy and specificity of the KRAS-targetting strategies and drugs [112].

### 6.3. Promising and Alternative Treatment Options Available for CRC

Some have proposed that the combination of gene therapy with standard treatments (chemotherapy and/or radiotherapy), immunotherapy, and stem cell therapy in the future are projected to warrant a cancer cure [70]. Despite the enthusiasm surrounding the targeted gene therapy, standard treatments such as chemotherapy, radiotherapy, and surgical resection still represent the gold standard methods for CRC management in many countries. 5-fluorouracil (5-FU), irinotecan, and oxaliplatin are among cytotoxic drugs that are commonly included in chemotherapy regiments, although adverse effects were recorded post-treatment in many CRC patients. Conventional methods are preferred due to cost efficiency, availability of drugs, and availability of experienced personnel, including oncologists, and technologies in hospitals and treatment facilities. Despite the wide application of chemotherapy, studies are currently being performed to improve this approach and reduce the side effects that may be experienced by the patients. For example, a treatment regimen with oxaliplatin in addition to fluoropyrimidine has been established as standard adjuvant chemotherapy since 2004 for patients with stage III CRC [117,118,119]. 

Genetic testing and mutational analysis helped to pave a way to more targeted treatment against many types of cancer including CRC. Targeted therapies in CRC also showed concrete positive results. For instance, better survival rates and responsiveness were recorded in metastatic CRC patients with the *BRAF* V600E mutation after a treatment consisting of the combination of encorafenib, cetuximab, and binimetinib [120]. This is a major outbreak in CRC therapy, as conventional chemotherapy for subjects with *BRAF* V600E mutations were not successful [121]. 

Moreover, recent breakthroughs in immunotherapy for the past decade have hailed it as the ‘Holy Grail’ for cancer treatment, including CRC. This was significantly verified by tumor profiling. The number and type of genomic changes have been proven to be essential to dictate the efficacy of immune checkpoint inhibitors, such as those involved in the PD-1/PDL-1 pathway [122,123]. Le et al. (2015) reported the clinical success of immune checkpoint blockage (ICB) by using pembrolizumab based on the mismatch-repair status in CRC patients [124]. This is supported by André et al. (2020), wherein a longer survivability was found in metastatic CRC subjects with MSI-H (microsatellite high) treated with Pembrolizumab compared to chemotherapy. Fewer adverse effects from the treatment also were found [119]. These findings solidified immunotherapy as the prime treatment for CRC. 

Some have argued that *KRAS* mutation is ‘undruggable’, as there is no clinically proven anti-*KRAS* therapy despite decades of research on this oncogene. However, the previous case report by Tran et al. (2016) exhibited an encouraging outcome whereby high effectiveness of antitumor therapy post-adoptive transfer of tumor-infiltrating CD8+ (expanded ex vivo) in a metastatic CRC patient with mutant *KRAS* G12D expression was demonstrated [125]. This personalized cell therapy approach in targeting mutant *KRAS* in colorectal tumors, utilizing the T-cell response, potentially will block the progression of various solid tumors. These reports suggested that CRC treatment using different modalities such as immunotherapy and targeted therapy may be the preferred options rather than gene therapy targeting p53 and KRAS mutations directly. 

### 6.4. Safety

As stated in the previous section, the more common method for gene therapy targeting p53 in cancer involves administrating wild-type p53 or suppressing p53 activity in defect cells. An obvious drawback of this approach concerns the overexpression of wild-type p53, which could induce adverse effects on normal cells [126,127].

A successful gene therapy also warrants the use of a safe vector. The safety issue of the delivery vehicle, such as an adenovirus, has been raised due to several reported incidences during clinical trials and due to a lack of understanding on its mechanism of action [128,129]. The United States Food and Drug Administration (USFDA) has disapproved the use of Introgen′s Advexin, which utilizes p53, due to safety concerns. Currently, there is no record of the Gendicine™ clinical data submission or approval status from the USFDA [130]. 

### 6.5. Ethical Concerns

Until 2017, nearly 2600 clinical trials involving gene therapy were completed, ongoing, or approved globally [131]. Ethically, gene therapy in germ cells has been controversial and, at the moment, is banned from human trials [132]. Additional ethical issues revolving around gene therapy include subject selection and consent, inadequate protection to subjects involved, and the question of bias or conflict of interest from the physician’s perspectives [133].

### 6.6. Cost-Effectiveness

Gene therapies are costly and this factor poses a great challenge in its implementation. For example, Zolgensma, a prescription gene therapy for spinal muscular atrophy (SMA), costs USD 2.1 million per patient, which ranks it as the most expensive drug in the world [134]. For cancer, Yescarta, which is a cell-based gene therapy used to treat Non-Hodgkin lymphoma patients, comes with a price of USD 0.373 million for a one-time dose [135]. The staggering prices of these approved therapies question the affordability of the treatment for patients, especially those from poor countries. 

The expensive expenditure in gene therapy originates from the personalized nature of this technology. Often the vectors and compounds are produced, tested, and manufactured on a smaller scale unlike those drugs in standard chemotherapy. It is worth noting that the development of a targeted gene therapy involves an extensive research and development (R&D) process, which adds to its overall cost and high per-patient price [136]. One also may also argue that a successful gene therapy administered at a single time may seem expensive but is cost-effective when compared to the costs of years of expensive and ongoing treatment for cancer patients. This highlights the definition of ‘values’ acquired from gene therapy, which lead to a better quality-of-life [137]. Gene therapy has been proven to improve body function, reduce suffering, and improve psychological stress due to the illness [137,138]. Efforts to establish a consensus is needed among the stakeholders, including policy makers and analysts, pharmaceutical companies, payers (both public and private), and key personnel from providers and organizations to further refine solutions and opportunities in regard to gene therapy access to patients [134]. 

## 7. Potential Outlook of Cancer Gene Therapy

A number of driver gene mutations, including of *TP53* and *KRAS* as stated in Table 1, resulted in immunogenic neoantigens. Among “hotspot” missense mutations of R175, G245, R248, R249, R273, and R282 in the *TP53* are found mutated in various types of cancers. These p53 missense mutations uniquely influence tumor behavior and promote carcinogenesis, thus providing insight into more targeted cancer treatment [139]. For instance, targeting p53 gene mutations of R175H and R273H has been reported to restore p53 mutants’ DNA binding activity and to abolish tumor development [140]. Additionally, a recent breakthrough discovery on significant anticancer properties of recently approved FDA drugs, namely Sotorasib (AMG 510), in subjects with solid tumors (non-small cell lung carcinoma, colon, etc.) harboring the *KRAS* p.G12C mutation has shed light on cancer gene therapy. Sotorasib acts as a covalent inhibitor that specifically and irreversibly extinguishes *KRAS*^G12C^ activity [141]. These encouraging discoveries indicate the prospect of cancer gene therapy as a cancer gene vaccine and immunotherapy targeting mutation sites of *TP53* and *KRAS*. 

Currently, the major immunotherapy types for cancer treatment include (a) monoclonal antibodies targeting specific antigens [142]; (b) ICB to reinvigorate T-cell immunity [143]; (c) CAR-T cell therapy [144]; (d) oncolytic viruses to eliminate specific cancer cells; and (e) cancer vaccines [145,146]. One of these strategies, namely therapeutic DNA cancer vaccines, may represent a very promising strategy to activate patients’ immune system to combat cancer [147,148]. It is worth noting that until now, pre-clinical and clinical trials indicated that therapeutic cancer vaccines on their own may not significantly improve cancer outcome. A DNA vaccine is proposed to be coupled with other personalized modes of treatment to be provided as a combined therapy after the analysis of a patient’s genetic profile and biomarkers that predict treatment response to a specific drug or agent. This will ensure better efficacy of the therapy and reduce the risk of adverse effects of the treatment [147]. 

## 8. Conclusions

There are currently significant advancements in gene therapy for CRC. Despite many promising results from pre-clinical studies, the efficacy of targeted gene therapy in oncology must be tested and further verified in large randomized clinical trials. Two of the major players in colorectal carcinogenesis, specifically *TP53* and *KRAS* genes, serve as robust targets for gene therapy. However, more fundamental studies are required to dissect the roles and regulations of p53 and KRAS pathways in tumorigenesis. Targeted gene therapy also needs to be scrutinized more carefully to ensure the best strategies for combating CRC. Currently, this approach is still far from the gold standard of CRC therapy, mainly due to its high cost and the availability of other treatment modalities such as immunotherapy that warrant promising results in CRC patients.

## Figures and Tables

**Figure 1 ijms-22-11941-f001:**
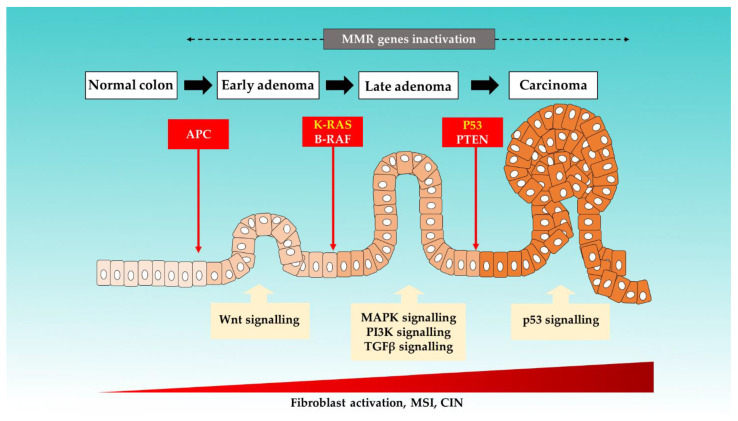
The proposed mechanisms of sporadic events for CRC progression from the normal colon.

**Figure 2 ijms-22-11941-f002:**
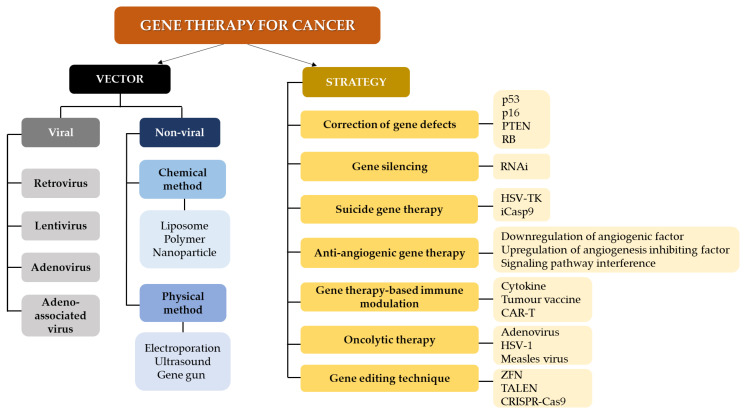
Gene therapy approaches and vectors for cancer treatment. Methods for gene therapy in cancer include RNA interference (RNAi), herpes simplex virus thymidine kinase (HSV-TK), inducible caspase 9; CAR-T (iCasp9): chimeric antigen receptor T-cell therapy (CAR-T), herpes simplex virus 1 (HSV-1); zinc finger nucleases (ZFN), transcription activator -like effector nucleases (TALEN) and clustered regularly interspaced short palindromic repeats-Cas9 (CRISPR-Cas9).

**Table 1 ijms-22-11941-t001:** Common genetic mutations in solid tumors.

Gene	Cancer Type and Origin				
	Breast	Colorectal	Cervix	Lung	Prostate
*APC*		●			
*ARID1A*			●		
*ATM*	●				●
*BMPR1A*		●			●
*BRAF*				●	●
*BRCA1*	●		●		
*BRCA2*	●				
*BRIP1*	●				
*CASP3*			●		
*CDH1*	●	●			
*CDKN2A*				●	
*CHEK2*	●				●
*DDR2*				●	
*EGFR*				●	
*EML4-ALK*				●	
*ERBB3*			●		
*FGFR1*				●	
*HER2*	●			●	
*KRAS*	●	●		●	●
*MET*				●	
*MLH1*		●			●
*MSH2*		●			●
*MSH6*		●			●
*MUTYH*		●			
*MYC*					●
*NTRK1*				●	
*PALB2*	●				●
*PIK3CA*					●
*PMS2*		●		●	●
*PTEN*	●	●	●	●	●
*RET*				●	
*RB1*					●
*ROS1*				●	
*SMAD4*		●		●	
*SHKBP1*			●	●	
*SOX2*				●	
*STK11*	●	●			
*TP53*	●	●	●	●	●
*TGFBR2*			●		

The dots represent genetic mutations that may present in the selected solid tumours.

**Table 2 ijms-22-11941-t002:** Viral and non-viral delivery approaches for CRC gene therapy.

Vector	Delivery Systems/Strategies	Findings	Refs
Non-viral	Local administration of pmiRNA-K-ras molecules into LoVo tumors by electroporation	MicroRNAs (miRNAs) targeting K-ras significantly downregulated *K-ras* expression and cell growth post *in vitro* electrotransfection	[73]
	Nanoparticles consisting of a core of high-molecular weight linear polyethylenimine (LPEI) complexed with DNA and enclosed by a shell of polyethyleneglycol-modified (PEGylated) low-molecular weight LPEI	Greater ratio of tumor to non-tumor transfection in intravenous delivery of the core/PEGylated shell (CPS) nanoparticles in comparison to conventional and commercially available *in vivo* gene delivery system	[74]
	Nano-sized cationic polymeric gene delivery system for glucose-6-phosphate dehydrogenase (G6PD) short hairpin RNA (shRNA) delivery	Greater oxaliplatin anti-tumor effects in cell-based xenografts and patient-derived xenografts (PDX)	[75]
	Self-assembling of DOTAP and MPEG-PLA (DMA), carrying interleukin 12 (IL-12) plasmid (DMP-pIL12 complex)	Significant inhibition of tumor growth in mouse model treated with the DMP-pIL12 complex (inhibition of angiogenesis and promoting both programmed cell death and a lower proliferation rate)	[76]
	PAMAM (G4 and G5) dendrimers modified by the alkyl-carboxylate chain, PEG, and cholesteryl chloroformate bearing TRAIL plasmid	PAMAM G4-alkyl-PEG (3%)-Chol (5%)-TRAIL complexes at C/P ratio 4 could significantly promote cell death compared to the unmodified PAMAM vector in both *in vitro* and *in vivo* conditions	[77]
	Peptide-functionalized hybrid delivery system	The encoding mRNA from the suicide gene Bim and a locally administered mBim/DMP-039 complex significantly inhibited proliferation in two colon cancer models	[78]
Viral	Replication-deficient adenovirus containing a bifunctional fusion gene: CD:uracil phosphoribosyltransferase (UPRT) (AdCDUPRT)	Treatment of intratumoral AdCDUPRT and intraperitoneal 5-FC in athymic mice with colon cancer xenografts significantly suppressed tumor growth in comparison to the untreated group, whereas no significant effect was found in AdCD/5-FC-treated mice	[79]
	An oncolytic virus of Onyx-015 administered to metastatic colorectal cancer patients by hepatic artery infusion	Majority of the subjects with stable disease at 3 months showed a unique radiographic pattern of transient tumor growth (10–48%) after the initial treatment of Onyx-015, followed by significant tumor necrosis and regression	[80]
	AAV expression vector pAM/CAG-WPRE.poly(A) cloned with the survivin mutant (Cys84Ala; Sur-Mut(Cys84Ala)) to generate recombinant AAV-Sur-Mut(Cys84Ala) virus	rAAV-Sur-Mut(Cys84Ala) promoted cell death and inhibited both angiogenesis and tumor growth	[81]
	AAV-mediated human interferon beta (IFN-beta) gene driven by the hTERT promoter	AAV2-IFN-β under the control of the hTERT promoter suppressed tumor growth (>90%) and increased survival of mice with CRC and lung cancer	[82]
	AAV-mediated survivin mutant Thr34Ala [rAAV-Sur-Mut(T34A)]	The treatment of recombinant AAV [rAAV-Sur-Mut(T34A)] significantly enhanced the anticancer activity of oxaliplatin and extended the survival of treated animals	[83]
	rAAV bearing four and a half LIM domains of protein 2 (FHL2)-shRNA	Treatment of rAAV-FHL2-shRNA produced significant anti-tumorigenic effects in nude mice and this activity was enhanced in combination with 5-FU treatment	[84]
	Human cathelicidin CAMP gene overexpressing the AAV (AAV2-cathelicidin)	Significant reduction in tumoral mouse collagen COL1A2 mRNA and protein expression in the azoxymethane/dextran sodium sulfate (AOM + DSS) mouse colonic tumor model post-intravenous administration of cathelicidin expressing AAV	[85]
	Full-length cetuximab antibody cloned into two serotypes of adenoviral vectors, termed as AdC68-CTB and Hu5-CTB	A single dose of AdC68-CTB or Hu5-CTB resulted in sustained expression of cetuximab and greatly inhibited tumor growth in NCI-H508 or DiFi-inoculated nude mice	[86]
	Enadenotucirev, chimeric adenovirus type 11p [Adp/adenovirus type 3 (Ad3)] virus	N/A (recruiting phase) Intervention of chemoradiation + enadenotucirev Phase I clinical trial (NCT03916510)	[87]
	Intra-tumoural injection or intravenous infusion of a Group B oncolytic adenovirus (ColoAd1) (Enadenotucirev) in patients with resectable tumors	N/A Phase 1 clinical trial (NCT02053220)	[88]

N/A: not available.

## Data Availability

Not applicable.

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
