# Peer review of "Gene Therapy Targeting p53 and KRAS for Colorectal Cancer Treatment: A Myth or the Way Forward?"

_ijms, 2021, doi:10.3390/ijms222111941_

Round 1

Reviewer 1 Report

Hasbullah submitted a manuscript entitled "Gene therapy targeting p53 and KRAS for colorectal cancer treatment: A Myth or The Way Forward" in IJMS. Although this manuscript summarized the advance and challenge in cancer gene therapy for colorectal cancer, this manuscript did not organize well.

  1. The Table 1 showed several driver genes mutation across different cancer type. But table 1 is too simple to tell whole story. The incidence of these driver mutation fraquence is quite different in different cancer. Besides, these mutations are not all suitable for gene therapy targets even in TP53 and KRAS. Although TP53 is highly mutated in CRC (~60%), the mutation sites in TP53 are diverse between patients, making it difficult as gene therapy targets.
  2. Part of these driver gene mutations results in immunogenic neoantigens. Maybe cancer gene therapy should be considered cancer gene vaccine and immunotherapy targeting these mutation sites.   Recent studies showed targeing TP53-R175H and KRAS-G12D are possible for the target for TCRmimic cancer therapy, suggesting these mutation sites can be considered as neoantigen for immunotherapeutic targets or cancer gene therapy by vaccination.
  3. Figure 2 is too simple. Several clinical trials showed the clinical response by targeting thes driver mutation with cancer gene therapy, AdV, CAR-T cell therapy and et al.  The author should summarize and compare these different delivery systme (viral and non-viral) as a table.
  4. Overall, this manuscript did not organize well.

Author Response

Dear Reviewer

Thank you for your comment. Please refer to the revised manuscript, changes are made according to the suggestions.

Reviewer 1

Hasbullah submitted a manuscript entitled "Gene therapy targeting p53 and KRAS for colorectal cancer treatment: A Myth or The Way Forward" in IJMS. Although this manuscript summarized the advance and challenge in cancer gene therapy for colorectal cancer, this manuscript did not organize well.

  1. The Table 1 showed several driver genes mutation across different cancer type. But table 1 is too simple to tell whole story. The incidence of these driver mutation fraquence is quite different in different cancer. Besides, these mutations are not all suitable for gene therapy targets even in TP53 and KRAS. Although TP53 is highly mutated in CRC (~60%), the mutation sites in TP53 are diverse between patients, making it difficult as gene therapy targets.

Reply: Additional paragraph in Page 4, starting line 131 (written in red)

  1. Part of these driver gene mutations results in immunogenic neoantigens. Maybe cancer gene therapy should be considered cancer gene vaccine and immunotherapy targeting these mutation sites.   Recent studies showed targeing TP53-R175H and KRAS-G12D are possible for the target for TCRmimic cancer therapy, suggesting these mutation sites can be considered as neoantigen for immunotherapeutic targets or cancer gene therapy by vaccination.

Reply: Additional paragraph in Page 13, starting line 472 (written in red)

  1. Figure 2 is too simple. Several clinical trials showed the clinical response by targeting these driver mutations with cancer gene therapy, AdV, CAR-T cell therapy and et al. The author should summarize and compare these different delivery systems (viral and non-viral) as a table.

Addition at page 8, starting line 264. Table 2 on viral and non-viral delivery system follows (written in red)

  1. Overall, this manuscript did not organize well.

Reply: Revisions are made as per comment 1 to 3

Reviewer 2 Report

In this review the Authors give a comprehensive overview of the potentials and challenges of gene therapy targeting p53 and KRAS for the treatment of CRC. In my opinion the topic is interesting and clearly written.

To improve their paper, I only suggest the Authors to add references to the table 1

Best regards

Marina De Rosa

Author Response

Dear Reviewer 2 (Dr Marina De Rosa)

Thank you for your comment. Addition has been made per suggestion.

Reviewer 2

In this review the Authors give a comprehensive overview of the potentials and challenges of gene therapy targeting p53 and KRAS for the treatment of CRC. In my opinion the topic is interesting and clearly written.

To improve their paper, I only suggest the Authors to add references to the table 1

Reply: References added (page 3: Table 1 summarizes main genetic mutations detected in several of the most commonly diagnosed solid tumours [35-39])

Best regards

Marina De Rosa

Thank you

Marahaini Musa

Round 2

Reviewer 1 Report

No further comment